# Control of the temperature signal in Antarctic proxies by snowfall dynamics

Aymeric P. M. Servettaz[1], Cécile Agosta[2], Christoph Kittel[3, 4], and Anaïs J. Orsi[5]

[1]Biogeochemistry Research Center, Japan Agency for Marine-Earth Science and Technology, Yokosuka, 237-0061, Japan
[2]Laboratoire des Sciences du Climat et de l'Environnement, LSCE/IPSL, CEA-CNRS-UVSQ, Université Paris-Saclay, Gif-sur-Yvette, 91190, France
[3]Department of Geography, UR SPHERES, University of Liège, Liège, Belgium
[4]Institut des Géosciences de l'Environnement, University Grenoble Alpes/CNRS/IRD/G-INP, Grenoble, 38000, France
[5]Department of Earth, Ocean and Atmospheric Sciences, University of British Columbia, Vancouver, V6T 1Z4, British Columbia, Canada

*Correspondence to*: Aymeric P. M. Servettaz (servettaza@jamstec.go.jp)

**Abstract.** Antarctica, the coldest and driest continent, is home to the largest ice sheet, whose mass is predominantly recharged by snowfall. A common feature of polar regions is the warming associated with snowfall, as moist oceanic air and cloud cover increase the surface temperature. Consequently, snow accumulated onto the ice sheet is deposited under unusually warm conditions. Here we use a polar-oriented regional atmospheric model to study the statistical difference between average and snowfall-weighted temperatures. During snowfall, the warm anomaly scales with snowfall amount, with strongest sensitivity at low accumulation sites. Heavier snowfall in winter contributes to decrease the annual snowfall-weighted temperature, but this effect is overwritten by the event-scale warming associated with precipitating atmospheric systems, which particularly contrast with the extremely cold conditions in winter. Consequently, the seasonal range of snowfall-weighted temperature is reduced by 20 %. On the other hand, annual snowfall-weighted temperature shows 80 % more interannual variability than annual temperature, due to irregularity of snowfall occurrence and their associated temperature anomaly. Disturbance in apparent annual temperature cycle and interannual variability have important consequences for the interpretation of water isotopes in precipitation, which are deposited with snowfall and commonly used for paleo-temperature reconstructions from ice cores.

**Highlights**

 – Snow precipitation events in Antarctica are associated with positive surface temperature anomalies that scale with snowfall rates.
 – Temperature during snowfall has a seasonal amplitude reduced by 20 % relative to the daily temperature.
 – Annually snowfall-weighted temperature shows 80 % more interannual variability than annual temperature.
 – Water isotopes reflect snowfall-weighted temperature and may be affected by such biases.

**1 Introduction**

Antarctica is the coldest and driest continent on earth, and almost entirely covered by ice. The surface temperature remains below freezing year-round over most of the continent, allowing the snow to accumulate and form the ice sheet, which is recharged primarily by snowfall. Precipitating atmospheric systems in polar regions are known to increase the surface temperature (Uotila et al., 2011). This is caused by atmospheric perturbations and clouds strongly disturbing the surface energy balance: while short-wave radiation is reduced during overcast weather, absorption of snow-emitted long-wave radiation and downward radiation from the cloud base cause the net radiative heat flux to warm the surface (Nardino and Georgiadis, 2003; Van Den Broeke et al., 2006). Snow accumulated onto the Antarctic ice sheet is mostly deposited under cloudy conditions, except for the higher parts of Antarctic Plateau where clear-sky precipitations form a large part of the total ice accumulation (Stenni et al., 2016; Fujita and Abe, 2006; Bromwich, 1988). Even for clear-sky precipitations, despite the absence of clouds, a vapor-rich atmosphere emits more long-wave radiation and warms the surface (Gallée and Gorodetskaya, 2010; Genthon et

al., 2013). Atmospheric perturbations can increase surface wind-speed (Argentini et al., 2014; Vignon et al., 2017; Baas et al., 2019). This weakens or breaks the surface temperature inversion, due to increased turbulent mixing incorporating warmer air
from the upper temperature inversion layer (Hirasawa et al., 2000; Vignon et al., 2019; Heinemann et al., 2019). In addition to modification of the local energy balance, synoptic systems effectively transport heat inland Antarctica (Carleton and Whalley, 1988; Giovinetto et al., 1992). Large atmospheric perturbations thus modify heat flux and surface temperature (Uotila et al., 2011), causing occasional surface warming exceeding 10°C (Ganeshan et al., 2022). Atmospheric rivers, which are extreme water vapor fluxes associated with some cyclones, are also associated with intense warming, particularly in winter (Wille et
al., 2021). Consequently, days with snow or ice accumulation in Antarctica are warmer than average.

Knowledge of temperature variability in Antarctica strongly relies on paleoclimate studies to extend the time period beyond the observation period of the satellite era (Jones et al., 2016). Of the temperature proxies, water stable isotopes in ice are the most used in paleoclimate studies in Antarctica (Stenni et al., 2017), due to the widely available base material and a good understanding of the fractionation processes associated with precipitation formation (Markle and Steig, 2022). Due to Rayleigh
distillation during transport of moisture to cold regions, water isotopes reflect the condensation temperature of precipitations (Dansgaard, 1964). However, the relationship between average temperature at a location and isotopes in the snow is altered by deposition dynamics of snowfall-borne water isotopes (Werner et al., 2000; Persson et al., 2011; Casado et al., 2020), post-deposition processes such as ablation-redeposition and sublimation-condensation cycles (Steen-Larsen et al., 2014; Touzeau et al., 2016; Stenni et al., 2016; Münch et al., 2017; Hughes et al., 2021), and the difference between condensation and surface
temperature (Buizert et al., 2021; Liu et al., 2023). Water isotope measurements characterize ice deposited during snowfall events, therefore $\delta^{18}O$ (following the $\delta$ notation as in Dansgaard, 1964) is thought to better correlate with snowfall-weighted temperature than average temperature (Stenni et al., 2016; Fujita and Abe, 2006), as shown in isotope-enabled models (Sturm et al., 2010). Differences between the snowfall-weighted temperature and average temperature remain poorly described. Characterizing these differences will thus help understand the signal recorded in water isotopes, and quantify the effects of
precipitation intermittency in Antarctic ice cores (Masson-Delmotte et al., 2011).

In Antarctica, a few large precipitation events can cause most of the inter-annual accumulation amount and variability (Turner et al., 2019). Consequently, these events control the temperature signal imprinted in the deposited snow isotopic composition. The few days when most of the snow is accumulated may not represent average conditions, because they are associated with temperature increase, and may occur anytime in the year, causing aliasing of the seasonal signal by irregular sampling.
Quantifying the temperature changes associated with snowfall could highlight key mechanisms for interpretation of water isotopes in snow and ice, and ultimately improve water-isotope-based temperature reconstructions. Although the different heat transport mechanisms are relatively well understood and included in current atmospheric models, a climatology of the warming associated with snowfall events has not been made so far.

Covariance of precipitation and temperature at synoptic and seasonal scales was shown to affect the isotope-temperature slope
by changing the temperature that can effectively be recorded in an ice core (Sime et al., 2008). Changes in recordable temperature may be linked to precipitation changes rather than temperature changes (Krinner et al., 2006). In addition,

intermittency of precipitation induces isotopic variability non-related to the temperature, especially important at inter-annual scale for the low accumulation East Antarctic plateau (Casado et al., 2020). Spatial and temporal changes of snowfall intermittency impact the recordable temperature (Sime et al., 2008), which is partly responsible for the spatial and temporal

variations in isotope-temperature slope values (Sime et al., 2009a, b; Klein et al., 2019). Sub-sampling the temperature signal by snowfall affects the recordable temperature in water-isotopes, but the extent of this effect, and its variability along the variety of precipitation regimes in the entire Antarctic continent, have been poorly characterized. Although post-deposition effects can further modify isotope-temperature slopes after deposition (Sime et al., 2011; Casado et al., 2018), understanding the temperature changes at time of deposition, related to snow precipitation, at different timescales and locations can explain

some of the spatial and temporal diversity of the slopes.

Here, we study the surface temperature changes associated with snowfall, aiming to understand the signal that a precipitation-based proxy would carry at time of its deposition. Using the polar-oriented regional atmospheric model MAR, extensively evaluated for its representation of Antarctic surface mass balance (Agosta et al., 2019) and temperature (Kittel et al., 2021), we compute snowfall-weighted statistics to evaluate the average bias and interannual variability of temperature across

Antarctica for the period 1979–2020. We focus our analysis on the quantification of temperature anomaly, defined as the difference to the daily climate normal temperature, and suggest possible effects on water isotopes.

## 2 Methods

### 2.1 Regional Atmospheric Model

We use the polar oriented regional atmospheric model MAR, in its version MARv 3.12 to assess the variability of snowfall

and temperature in Antarctica. This model has shown good performance in its version 3.11 to represent the surface mass balance (defined as difference of total precipitation minus sublimation and runoff, Agosta et al., 2019), temperature (Kittel et al., 2021) and cloud (Kittel et al., 2022) variability. The version 3.12 used here differs mainly in the temperature transition of rainfall to snowfall where both can now co-occur in the model, and a correction of snowpack temperature (Antwerpen et al., 2022) as well as albedo tuning for the dense Antarctic snow. The simulation domain is on an Antarctic polar stereographic

projection (EPSG:3031), with horizontal resolution of 35×35 km. The MAR model is nudged to ERA5, the latest re-analysis product of the European Center for Medium-Range Weather Forecast, with the most accurate representation of both temperature and surface mass balance in Antarctica among climate re-analyses (Gossart et al., 2019). The use of the MAR regional model enables further detail in the near-surface atmospheric layers and spatial scale compared to ERA5. The simulation used in this study covers the 1979–2020 period (42 years). There is a notable lack of direct snowfall observations

in the low-accumulation regions of Antarctica, which hinders our ability to directly evaluate the model on this parameter. Modelled surface mass balance can be evaluated against observed accumulation, but includes several processes from snowfall to snow drift, evaporation of falling snow, and evaporation-condensation on the snow surface, which each come with their uncertainties. We evaluate the performance of MARv3.12 to represent temperature and surface mass balance in Figs. A1–A4.

## 2.2 Temperature averaging, difference, anomaly definitions and notations

Although the temperature recorded in water isotopes is imprinted at the condensation level (Jouzel and Merlivat, 1984), we chose to use 2-m air temperature for simplicity, because condensation levels change both spatially and temporally. Studies using water isotopes usually bypass the condensation to surface temperature changes by directly calibrating the isotope-temperature slope with 2-m temperature in most cases (e.g., Jouzel et al., 2007; Stenni et al., 2017), or applying a ratio of temperature changes that would be amplified at the surface (e.g., Jouzel et al., 2003). If we used the condensation-level
temperature, the difference with climate normal would depend on the level of precipitation formation, and may be vertically spread on the atmospheric column, making the comparison more complex. With condensation temperature, we would expect weaker seasonal cycles because winter surface cooling is amplified by a strong inversion, but long-term temperature variability may not change much as implied by deglaciation simulations (Liu et al., 2023). Choosing the 2-m temperature also enables comparison with available observations, and this is the level also considered in many paleotemperature reconstructions.

In this study, we use average temperature, calculated with the arithmetic mean:

$$T = \frac{\sum_{day=1}^{n_{days}} T_{day}}{n_{days}} \tag{1}$$

and snowfall-weighted temperatures, defined as the weighted average of temperature with daily snowfall rate as the weighting coefficient:

$$T_w = \frac{\sum_{day=1}^{n_{days}} T_{day} \times SF_{day}}{\sum_{day=1}^{n_{days}} SF_{day}} \tag{2}$$

In both cases, $T_{day}$ refers to the temperature on a given day, and $SF_{day}$ is the snowfall on the same day. Temperature averages can be computed for the entire study period ($n_{days} = 15341$), a year ($n_{days} = 366$), or on a given day of year ($n_{days} = 42$, used for climate normals $\overline{T}$ and $\overline{T_w}$). We define the climate normal temperature for each model grid cell as the average on the same day of year for the 42 years, and subsequently apply a 30-day rolling mean to smooth the signal.

To quantify the difference of temperature associated with snowfall, we define the snowfall-weighted temperature difference
as:

$$\Delta T = T_w - T \tag{3}$$

This metric has been previously described as precipitation-weighted biasing in Sime et al. (2008), although we chose not to name it bias to avoid the confusion with the modelling temperature bias, referring here to the difference in modelled vs observed temperature.

Decomposition of daily temperature $T_{day}$ into climate normal temperature on that day ($\overline{T}_{day}$) and daily anomaly to climate normal ($T'_{day}$) allows us to separate the seasonal and non-seasonal effects of snowfall weighting.

$$\Delta T = \frac{\sum_{day=1}^{n_{days}}\left(\bar{T}_{day} + T'_{day}\right) \times SF_{day}}{\sum_{day=1}^{n_{days}} SF_{day}} - \frac{\sum_{day=1}^{n_{days}}\left(\bar{T}_{day} + T'_{day}\right)}{n_{days}} \tag{4}$$

$$\Delta T = \underbrace{\frac{\sum_{day=1}^{n_{days}} \bar{T}_{day} \times SF_{day}}{\sum_{day=1}^{n_{days}} SF_{day}} - \frac{\sum_{day=1}^{n_{days}} \bar{T}_{day}}{n_{days}}}_{seasonal} + \underbrace{\frac{\sum_{day=1}^{n_{days}} T'_{day} \times SF_{day}}{\sum_{day=1}^{n_{days}} SF_{day}} - \underbrace{\frac{\sum_{day=1}^{n_{days}} T'_{day}}{n_{days}}}_{= 0}}_{non-seasonal} \tag{5}$$

A summary of abbreviations used is given in Table 1.

## 3 Results and discussion

### 3.1 Average temperature during snowfall

The temperature during snowfall is statistically higher than average temperature on the same day, as shown by the mostly positive temperature anomalies (Fig. 1). Despite a wide distribution of temperature anomalies at any given snowfall rate, the

average temperature anomaly increases with snowfall. Over Antarctica, there is a +5°C increase between snowfall rates of 0 (no snowfall) to 1 kg m$^{-2}$ day$^{-1}$, and a gradual increase of another +5°C as snowfall rates increase from 1 up to 100 kg m$^{-2}$ day$^{-1}$. Conversely, days without snowfall are 2°C cooler than average. For major regions of Antarctica, similar patterns are modelled with negative temperature anomalies on days without snowfall, and increasing temperature anomalies, up to +10°C for snowfall exceeding 50 kg m$^{-2}$ day$^{-1}$. The main difference for East Antarctic high elevations (Fig. 1f) is that the temperature

anomalies reach the +5°C threshold for snowfall rates of less than 1 kg m$^{-2}$ day$^{-1}$.

The positive anomaly associated with snowfall affects all the Antarctic continent although with varying intensity as shown by the map of differences between snowfall-weighted temperature and average temperature (noted $\Delta T$, Fig. 2, abbreviations listed in Table 1). Over the entire Antarctic continent and ice shelves, $\Delta T$ averages 5.4°C. While coastal regions and West Antarctica show $\Delta T$ of 0 to 5°C, the East Antarctic Plateau and ice shelves reach $\Delta T$ of up to 10°C. Interestingly, $\Delta T$ is the highest in large

topographical depressions such as in the Recovery (20° W, 80° S), Aurora (115° E, 75° S) and Wilkes (150° E, 70° S) basins, or the Byrd glacier catchment inland of transantarctic mountains (150° E, 80° S). On the other hand, over steep slopes and ridges, snowfall-weighted temperature is relatively closer to average temperature. Note that hoar frost is computed separately from snowfall in the model, and occurs in cold conditions (Schlosser et al., 2016). Therefore, ice accumulated by hoar frost can mitigate the warm conditions associated with snowfall, but this is not depicted by $\Delta T$ that accounts only for snowfall.

Another modelling study by Sime et al. (2008) showed $\Delta T$ of up to 10°C in East Antarctica for the present day, and lower values of about 5°C in west Antarctica, consistently with the results presented here. Our results mostly differ the coastal regions, and may relate to the increased resolution used in this study, or difference in modelling the physical processes of the katabatic-affected Antarctic slopes. In this work we focus on the quantitative temperature increase, but degradation of the climatic signal due to loss of correlation induced by precipitation intermittency has been treated in similar studies (Sime et al.,

2011; Casado et al., 2020).

To better understand the temperature anomaly associated with snowfall at the Antarctic scale, we analyse 10 sites where the impact of extreme precipitation events on total accumulation has previously been discussed (Turner et al., 2019), and show the temperature−snowfall relationship at these locations (inserts in Fig. 2). There are strong differences between sites located near the coast (Law Dome, East and West Peninsula, Gomez) compared to high elevation sites on the East Antarctic Plateau (High Plateau, Dronning Maud Land, Dome C): high elevation sites are characterized by low snowfall rates, but events causing snowfall larger than 1 kg m$^{-2}$ d$^{-1}$ are accompanied by a temperature increase of more than 10°C on average, and commonly up to 20°C (Table 2). These high elevation sites with low snowfall and large temperature anomalies are responsible for the sharp increase in temperature associated with low snowfall rates (Fig. 1f). Most locations reach temperature anomalies close to 10°C at their respective maximum snowfall (WAIS Divide, Gomez, Law Dome), except for sites where dry warming usually occurs, driven by Foehn (East Peninsula) or katabatic adiabatic compression (coastal slopes). Each site shows an increasing temperature trend with snowfall rate, with steeper slopes for sites with low accumulation (Fig. 2). Overall, days with snowfall are statistically warmer than average conditions, and increasingly so for higher snowfall rates at a given location.

The analysis of yearly snowfall-weighted temperature ($^{y}T_w$) and "true" yearly temperature ($^{y}T$, Fig. 3) further supports that the effect of snowfall weighting is not constant, and may differ along local parameters including the temperature, but also probably the precipitation regimes. Importantly, $^{y}T_w$ is not linear with $^{y}T$, suggesting that changes in the annual temperature are not matched by proportional changes in the snowfall-weighted temperature. This relationship may also change whether we average annually or at other time resolutions. Besides, any given $^{y}T$ is matched by a large distribution of $^{y}T_w$, which means that snowfall weighting induces variability in the temperature.

## 3.2 Variability of temperature during snowfall

Snowfall events, in particular large precipitation events, are an important source of variability in the Antarctic climate (Turner et al., 2019). As there is a clear link between snowfall intensity and temperature anomaly, the variability of snowfall translates to the variability of temperature. We first investigate the seasonality of temperature anomalies associated with snowfall (Fig. 4), by considering the climate normal snowfall-weighted temperature ($\overline{T_w}$, see Sect. 2 Methods for details on computation) and the climate normal temperature ($\overline{T}$). $\overline{T_w}$ differs from $\overline{T}$ by 3°C in summer, and up to 8°C in winter on average on the Antarctic ice sheet. The larger difference in winter results from the attenuation of near-surface temperature inversion during the passage of precipitating atmospheric systems. Indeed, $\overline{T}$ reaches extremely low temperatures in winter, driven by the strong surface radiative cooling (Connolley, 1996; Hudson and Brandt, 2005; Genthon et al., 2021). While it contributes to large variability in winter temperatures compared to summer season (Ricaud et al., 2017), snowfall consistently occurs under warm conditions. The seasonal amplitude of $\overline{T_w}$ is thus 20 % lower than that of $\overline{T}$ on average in Antarctica (Fig. 5). Reduction of seasonal amplitude occurs consistently over the Antarctic continent, and is strongest in coastal slopes.

In winter, cyclogenesis is slightly higher (Uotila et al., 2011), and atmospheric blockings are more frequent (Wille et al., 2021; Scarchilli et al., 2011), increasing the probability of poleward moisture transport and Antarctic snowfall. This results in higher

snowfall in the winter months at the Antarctic scale (Fig. 4b) and causes snowfall-weighted temperature to be influenced more by the winter season, when the snowfall-related warming is the strongest.

We decompose the contributions of seasonal distribution of snowfall (Fig. 6a) and event-related daily temperature anomaly (Fig. 6b) to $\Delta T$. As most of Antarctica receives more snowfall in winter (Palerme et al., 2017), the difference induced by seasonality averages −0.7°C over the ice sheet, but rarely exceeds −3°C. On the contrary, snowfall event-related warming causes a difference of +6.1°C, and dominates the difference between snowfall-weighted and all-day temperatures. These results are also in good agreement with the frequency decomposition of Sime et al. (2008), who showed that most of $\Delta T$ signal was

in the synoptic signal, comparable to daily anomaly of temperature used here. Although the seasonal signal is mostly negative in Fig. 6a, we note weakly positive $\Delta T$ in Victoria Land, where Sime et al. (2008) also showed positive $\Delta T$ for their seasonally band passed signal. The extent of this positive region is greater in Sime et al. (2008), extending well within continental East Antarctica, but may be related to the discrepancy in modelled seasonal precipitation for the dry East Antarctic plateau, with a summer precipitation maximum causing positive $\Delta T$ in Sime et al. (2008) as opposed to the winter maximum causing negative

$\Delta T$ here (Figs. 5 and 6, High Plateau site). In another study using the same method, Masson-Delmotte et al. (2011) find much stronger $\Delta T$ over the East Antarctic plateau, linked to seasonal effects on temperature. However, this difference is likely to emerge from the ERA40 re-analysis used, which was documented with a lack of winter precipitation and cyclone intensity in winter in the driest regions of Antarctica (Bromwich et al., 2007; Marshall, 2009), which leads to unrealistically large seasonal effects of precipitation weighting.

The dampened seasonal amplitude of snowfall-weighted temperature results from averaging 42 years, smoothing out interannual variability. While yearly averaged temperature ($^yT$) is relatively stable through time (Fig. 7a), interannual snowfall is highly variable, especially for the winter season (Casado et al., 2020; Turner et al., 2019). It causes the yearly snowfall-weighted temperature ($^yT_w$) to vary significantly from one year to another, with a standard deviation increased by +80 % on average over the ice sheet compared to $^yT$ (Fig. 7d). The variability is especially increased in Dronning Maud Land and the

Eastern part of West Antarctica, facing the Ronne Ice Shelf, where the interannual variability of $^yT_w$ can be 200 % larger than $^yT$ variability. Previous studies highlighted that the large variability of winter temperatures causes the winter season to be dominant in the interannual temperature variability, as a warm year is usually due to a warm winter, often accompanied with one or multiple snowfall events in winter (Persson et al., 2011; Casado et al., 2020; Servettaz et al., 2020). Despite the reduced seasonality of snowfall-weighted temperature and the tendency to oversample warm winters, its interannual variability is

increased. In other words, the temperature averaged equally over all days of a year is more stable than the temperature taken during snowfall only, because of the sporadic nature of snowfall, which subsamples the temperature of a limited number of days (Fujita and Abe, 2006; Turner et al., 2019) at random times of the year, and with a temperature bias which depends on precipitation intensity. On the interannual scale, the variability of yearly averaged temperature is thus enhanced when weighting with snowfall.

## 3.3 Implications for water isotopes

Water isotopes are used in Antarctic paleoclimate studies as a proxy for temperature due to the relationship between air temperature at condensation and the isotopic ratio in precipitation (Stenni et al., 2017; Dansgaard, 1964). Ice cores retrieve material accumulated over time onto the ice sheet, where the ice mass contribution depends on the snowfall. Therefore, snowfall-weighted temperatures provide an analogue to the isotopes we can expect to measure in an ice core. We thus discuss how temperature during snowfall may affect $\delta^{18}O$ at time of snow deposition, prior to post-deposition effects that occur later and further modify snow isotopes. Although we discuss mainly water isotopes due to their preponderance in paleo-temperature studies in Antarctica, these effects would theoretically apply to any snowfall-dependent temperature proxy.

Water isotopes in snow are deposited under warmer-than-normal conditions (Fig. 2), which leads to higher-than-expected $\delta^{18}O$. Some climatic information is lost for $\delta^{18}O$ as cold days are not recorded, or recorded with lower weight. Previous works suggested that $\delta^{18}O$ could be seasonally biased due to the annual cycle of snowfall (Markle and Steig, 2022; Werner et al., 2000; Persson et al., 2011), but here we showed that the temperature increase associated with precipitation events is clearly the main factor controlling snowfall-weighted temperature in Antarctica. The variety of $\Delta T$ across modern Antarctica suggests that it depends on precipitation regimes and amplitude of temperature change during precipitation at a given site. The stability of $\Delta T$ through time is thus decisive for temperature reconstructions based on isotopes because temporal changes of $\Delta T$ could accompany temperature and precipitations changes, and hinder the $\delta^{18}O$-based reconstructions. Here, the short study period does not allow to evaluate temporal changes in $\Delta T$, but such changes may be responsible for modifications of $\delta^{18}O$–temperature slopes at longer timescales, as was suggested for the glacial-interglacial range (Buizert et al., 2021). Previous studies also highlighted that despite being weaker that non-seasonal effects in absolute value, seasonal effects on $\Delta T$ are the more likely to vary with climate as the seasonality of precipitation changes (Sime et al., 2008), in response to sea ice and moisture source changes (Holloway et al., 2016). Given the spatial variability of $\Delta T$, we advise against the use of spatial gradients to define isotope-temperature slopes for temporal reconstructions.

Moreover, the reduced annual cycle of $\overline{T_w}$ relative to $\bar{T}$ may reflect a lower annual cycle of $\delta^{18}O$, which can explain why seasonal $\delta^{18}O$-temperature slope appears lower in precipitation $\delta^{18}O$ studies than in simple isotopic models (Casado et al., 2018). On the other hand, averaging temperature and isotopes yearly to define the $\delta^{18}O$-temperature slope may increase the slope value because of the higher interannual variability of $\delta^{18}O$ induced by irregularity of the snowfall distribution, similarly to snowfall-weighted temperature. Links were found between Antarctic temperature and large-scale atmospheric circulation patterns in the Southern Hemisphere such as the southern annular mode (abbreviated SAM, Marshall and Thompson, 2016), possibly influencing the $\delta^{18}O$ (Abram et al., 2014; Kino et al., 2021). Nevertheless, we did not find any significant correlation between the SAM and yearly or monthly snowfall-weighted temperature difference. Detecting possible links between the SAM, or other climate modes, and the precipitation-weighted temperature (or $\delta^{18}O$) would require a more detailed investigation, and may be explored in a different study.

Slopes of snowfall-weighted $\delta^{18}O$ of precipitations against snowfall-weighted temperature have been previously suggested (Fujita and Abe, 2006; Sturm et al., 2010; Servettaz et al., 2020). Precipitation-weighted $\delta^{18}O$ makes physical sense because it mimics ice core signal — omitting the post-deposition effects —, but paleo-climate reconstructions seek temperature rather than snowfall-weighted temperature. This work highlights the critical importance of event-related warming on the temperature during snowfall, which reduces seasonal amplitude, while irregular snowfall distribution enhances interannual variability of the temperature possibly recorded in water isotopes. This explains at least partly a higher interannual variability of precipitation-weighted $\delta^{18}O$, causing increased $\delta^{18}O$-temperature slope in most of Antarctica at interannual scale compared to seasonal scale (Goursaud et al., 2018), and low correlations between modelled $\delta^{18}O$ and temperature at annual scale (Münch et al., 2021). Simulation of $\delta^{18}O$ signals that would be recorded in Antarctic Peninsula ice cores also revealed that the interannual variability in $\delta^{18}O$ may show poor correlation to temperature variability even in high accumulation regions (Sime et al., 2009b). Non-linearities in the snowfall-weighted temperature as temperature and climate changes (Fig. 3) may be responsible for non-linear response of isotopes to temperature and underestimation of temperature maximum in warm periods, through increased winter precipitations (Sime et al., 2009a).

Understanding the effect of snowfall weighting on temperature average at ice coring sites will help reconstruct paleo-temperature more accurately. Depending on the targeted time frame for temperature reconstruction from isotopes, be it seasonal (Jones et al., 2023) or pluriannual (Stenni et al., 2017), the reconstructed temperature range may be lessened or increased. Moreover, using slopes variable through time would result in better temperature quantification, because the slope depends on the temperature range and the location (Sime et al., 2009a), and may vary through time (Klein et al., 2019).

Quantifying the local effect of snowfall-weighting on temperature range can help refine the temperature-isotope slopes for a more accurate estimation, and it should be done for different settings from glacial to warmer-than-present interglacial climate. Future temperature reconstructions could consider proceeding in two steps: (1) determine the snowfall-weighted temperature from water isotopes, for which the correlation is generally good and can be determined by Rayleigh-type models (e.g., Markle and Steig, 2022), then (2) determine the average (non-weighted) temperature through site-calibrated $T_w - T$ slope, calculated for the matching temporal resolution (similarly to Fig. 3, but here we only show the $^yT_w - {}^yT$ slope computed with yearly averages, and include all of Antarctica), while accounting for the difference in temperature between condensation level and surface, often dictated by inversion strength. Greater snowfall-weighted temperature differences at low-accumulation sites suggest that changes in snowfall regimes could impact the temperature difference, and thus bias the reconstructions from isotopes. Further work is necessary to fully understand how change in snowfall dynamics may influence temperature reconstructions from isotopes, which may be facilitated by atmospheric models equipped with isotopes.

## 4 Conclusions

We evaluated the temperature during snowfall in Antarctica using the regional atmospheric climate model MAR. Positive temperature anomalies usually accompany snowfalls, and the anomalies tend to increase with snowfall rate at any given site.

The slope of temperature increase as a function of snowfall rate is strongest at sites with low accumulation, so that even locations with low snowfall rates experience a strong snowfall-weighted temperature difference. Over the Antarctic continent, this difference averages 5.4°C. Temperature anomalies are typically strongest in winter, which leads to a 20 % reduced amplitude in the seasonal cycle of temperature during snowfall. Larger temperature anomalies during winter also offset the slightly higher seasonal contribution of winter precipitations which would reduce the snowfall-weighted temperature by 0.7°C. Year to year irregularities in snowfall distribution contribute to randomly subsample temperature, and increase interannual variability of snowfall-weighted temperature, making it 80 % more variable. Under the assumption that water isotopes reflect snowfall-weighted temperature, these biases will transfer to the isotopic signal in ice cores, which may explain the necessity to adjust isotope-temperature slope values depending on the timeframe of the reconstruction. Non-linearities in the snowfall-weighted temperature compared to site temperature confirm previous results that using linear isotope-temperature slopes may lead to overestimate temperature decrease in cold periods and underestimate temperature increase in warm periods. While we focused on the 1979–2020 period in this study, potential changes in precipitation regimes at longer timescales may be associated with changes in the snowfall-weighted biases, and deserve attention in future studies to adjust the isotope-temperature slopes accordingly for quantitatively accurate paleo-temperature reconstructions.

## Appendix A Evaluation of MAR performance

(Figures only)

## Code availability

The code for the regional atmospheric climate model MAR (Modèle Atmosphérique Régional) is available upon registration at https://gitlab.com/Mar-Group/MARv3 [last access: August 2023]. Python code for computations and figure creation is available at https://gitlab.com/aymericservettaz/antarctic-temperature-anomalies [last updated: October 2023].

## Data availability

The MAR simulation used in this study is available at doi.org/10.5281/zenodo.8408864 [last updated: October 2023, under embargo until acceptation of the present manuscript].

## Author contribution

CA and CK participated in the development of the MAR model and its application in the Antarctic region. AS designed the study, performed formal analysis of model outputs, and wrote the initial draft. AS, CA and CK created figures. AO provided ideas and guidance for the manuscript composition. All authors contributed to writing and editing the manuscript.

## Competing interests

The authors declare that they have no conflict of interest.

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

**Tables**

**Table 1.** Abbreviations used in this study.

| Abbr. | Full Name | Mathematic Definition | number of values at each location |
|---|---|---|---|
| $T_w$ | snowfall-weighted average temperature | $T_w = \dfrac{\sum_{day=1}^{n_{days}} T_{day} \times SF_{day}}{\sum_{day=1}^{n_{days}} SF_{day}}$, for the entire study period | 1 |
| $T$ | average temperature | $T = \dfrac{\sum_{day=1}^{n_{days}} T_{day}}{n_{days}}$, for the entire study period | 1 |
| $^{y}T_w$ | yearly snowfall-weighted average temperature | $^{y}T_w = \dfrac{\sum_{day=1}^{n_{days}} T_{day} \times SF_{day}}{\sum_{day=1}^{n_{days}} SF_{day}}$, for one year | 42 |
| $^{y}T$ | yearly average temperature | $^{y}T = \dfrac{\sum_{day=1}^{n_{days}} T_{day}}{n_{days}}$, for one year | 42 |
| $\Delta T$ | snowfall-weighted temperature difference | $\Delta T = T_w - T$ | 1 |
| $\overline{T_w}$ | climate normal snowfall-weighted temperature | For each day of year, same as $T_w$ then 30-day rolling average | 366 (one per day of year) |
| $\bar{T}$ | climate normal temperature | For each day of year, same as $T$ then 30-day rolling average | 366 (one per day of year) |
| $T'$ | Daily temperature anomaly to climate normal | Daily difference to $\bar{T}$ on the corresponding day of year | One per day |

**Table 2.** Values of temperature anomalies (T') for snowfall rates higher than 1 kg m$^{-2}$ d$^{-1}$. Locations of sites are shown in Fig. 2. For each site, average (arithmetic) and quantiles for different percentages are shown.

| T' (°C) for SF > 1 kg m$^{-2}$ d$^{-1}$ | Dronning Maud Land | High Plateau | Law Dome | Dome C | Ross Ice Shelf | Ocean | WAIS Divide | Gomez | West Peninsula | East Peninsula |
|---|---|---|---|---|---|---|---|---|---|---|
| average | 13.6 | 23.2 | 4.1 | 20.3 | 10.3 | 3.7 | 7.2 | 2.9 | 1.3 | 1.5 |
| q95 | 20.8 | 27.9 | 10.6 | 31.3 | 22.0 | 13.4 | 14.4 | 9.8 | 7.3 | 8.9 |
| q84 (+1σ) | 18.1 | 27.2 | 8.0 | 26.9 | 16.9 | 9.2 | 11.8 | 7.3 | 4.9 | 5.3 |
| q50 | 13.7 | 25.2 | 3.9 | 19.9 | 9.3 | 2.6 | 7.1 | 2.9 | 1.3 | 1.2 |
| q16 (-1σ) | 8.9 | 19.1 | 0.5 | 12.8 | 4.2 | -0.4 | 2.9 | -1.1 | -1.9 | -2.1 |
| q05 | 5.6 | 17.2 | -1.8 | 10.2 | 1.6 | -4.0 | 0.0 | -4.3 | -4.7 | -4.4 |

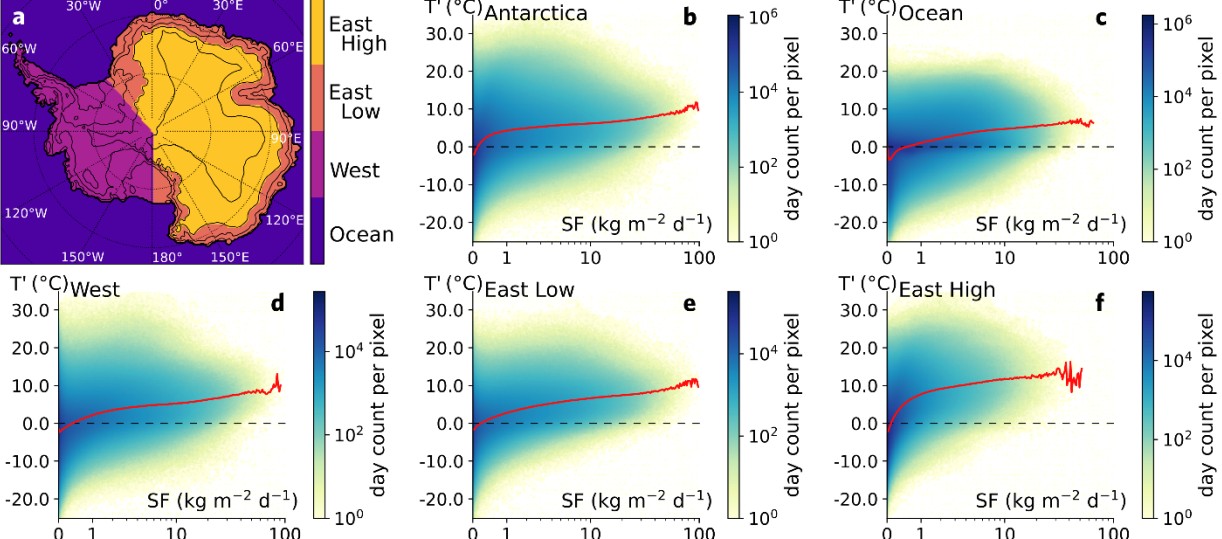

**Figure 1.** Scatter heatmaps of the daily temperature anomaly to climate normal as a function of daily snowfall, for different major regions of Antarctica, for the period 1979–2020. The temperature anomaly $T'$ is defined as the difference between daily 2-m temperature and the climate normal $\bar{T}$ (temperature on the average seasonal cycle for this day, see Sect. 2 methods) on the corresponding model cell, for each day. Scatter heatmaps are represented for each region defined in (**a**), with Antarctica (**b**) regrouping every point on the surface of the Antarctic ice sheet (including ice shelves). Ocean (**c**) is the remaining model domain in the Southern Ocean. Antarctica is further subdivided in West (**d**) for longitudes between 180° W and 45° W, East Low (**e**) for longitudes between 40° W and 180° E and elevation below 2000 m and East High (**f**) for longitudes between 45° W and 180° E and elevation above 2000 m. For heatmaps (**b–f**), the red line represents the average temperature anomaly given the snowfall rate, dashed lines highlight the T'=0 (no anomaly), x-axis is linear with $\log_{10}(1+\text{snowfall})$, and scatter density was counted as the number of days in each pixel-sized bin after projection on the logarithmic snowfall scale.

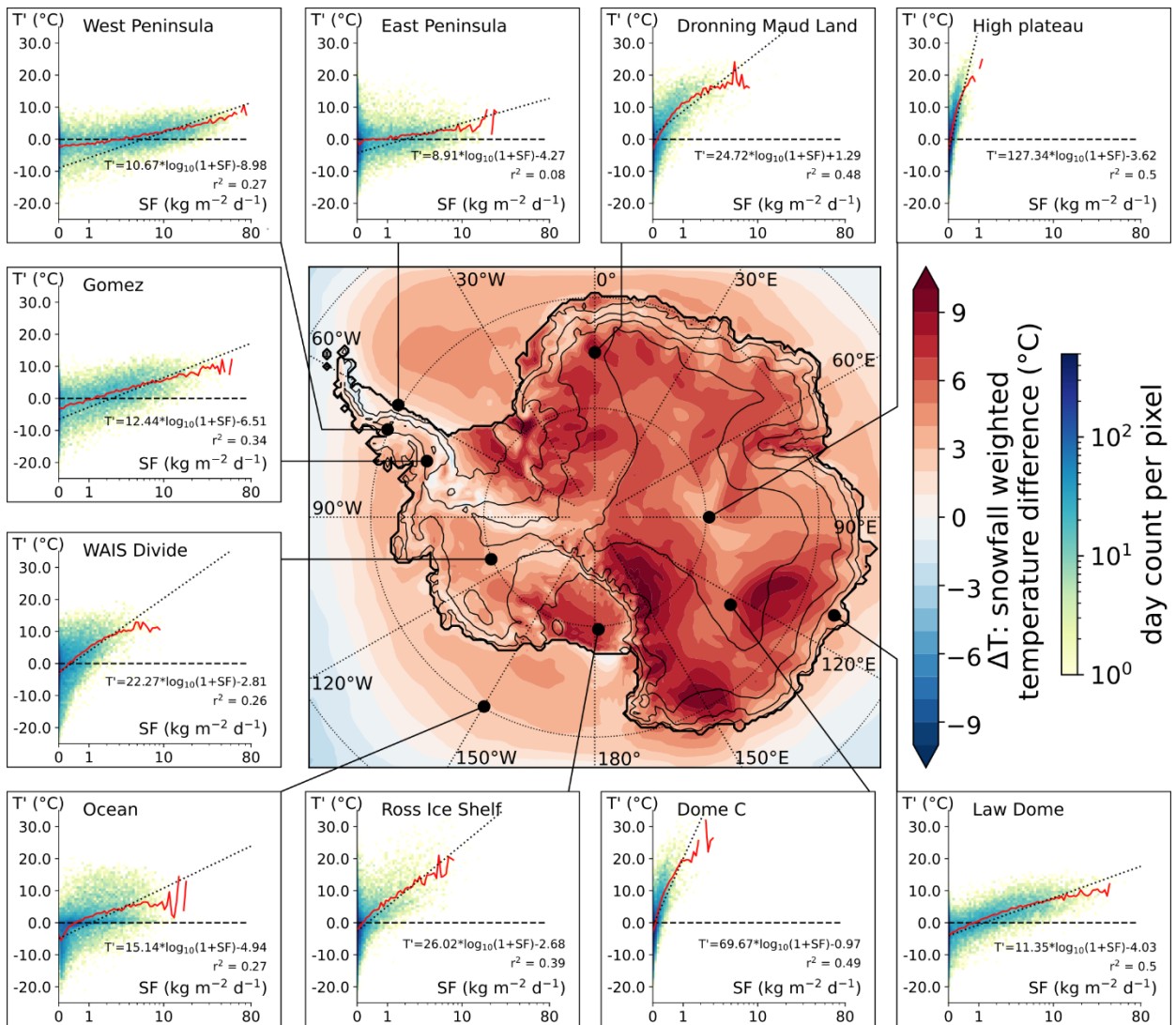

**Figure 2.** Map of snowfall-weighted 2-m temperature differences ($\Delta T$), and scatter heatmaps of site-specific temperature anomalies (T')
against snowfall rates (SF). Thick black lines indicate the extent of the Antarctic continent (including ice shelves). Thin lines show the 100
555   m, 1000 m, 2000 m, and 3000 m elevation contours. Black dots show the location of 10 selected locations where the scatter density heatmap
of temperature anomaly to climate normal is shown in each insert (same as Fig. 1, but for a single model cell). Dotted lines represent linear
trends computed for each insert on the $\log_{10}(1+SF)$ scale using snowfall-weighting coefficients (following the regression method described
in Servettaz et al., 2020). Days without snowfall or with extremely low snowfall rates, below 0.05 kg m$^{-2}$ d$^{-1}$, were excluded from the trend
computation. All trends are significant (p value < 0.01). T' – Temperature anomaly to climate normal; SF – Snowfall.

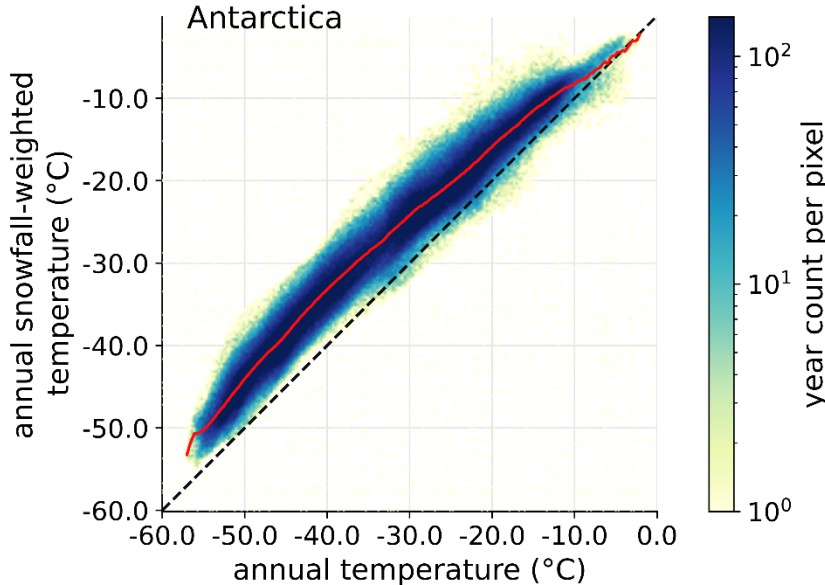

**Figure 3.** Scatter heatmap of annual snowfall-weighted temperature ($T_w$) as a function of annual temperature ($T$) for every model point on the surface of the Antarctic ice sheet (including ice shelves). The red continuous line represents the average snowfall-weighted temperature given the annual temperature, dashed line highlights 1:1 line.

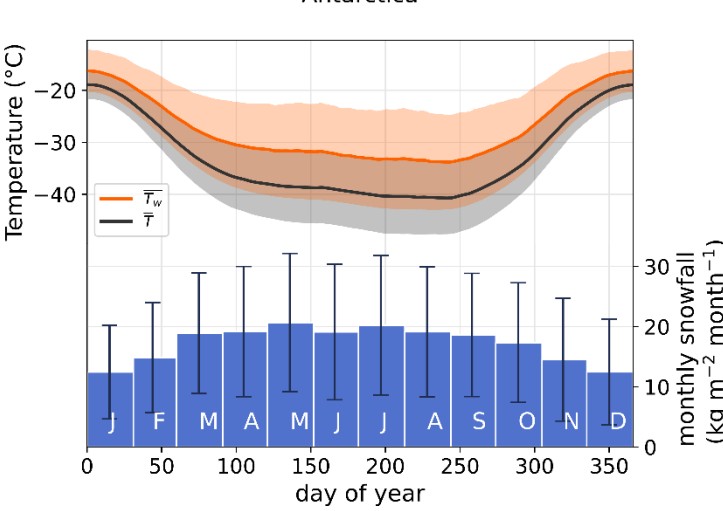

 **Figure 4.** Seasonal cycles of temperature and snowfall averaged over Antarctica. (**a**) climate normals of temperature ($\overline{T}$) and snowfall-weighted temperature ($\overline{T_w}$). Shadings indicate 1σ standard deviation. (**b**) monthly snowfall, with 1σ standard deviation indicated by error-bars. For both panels we included all model points on the Antarctic continent, including ice shelves. Climate normal temperatures were computed as arithmetic (for $\overline{T}$) or snowfall-weighted (for $\overline{T_w}$) average of the same day of year for all the 1979–2020 period, then smoothed with a 30-days rolling mean (see Sect. 2 Methods).

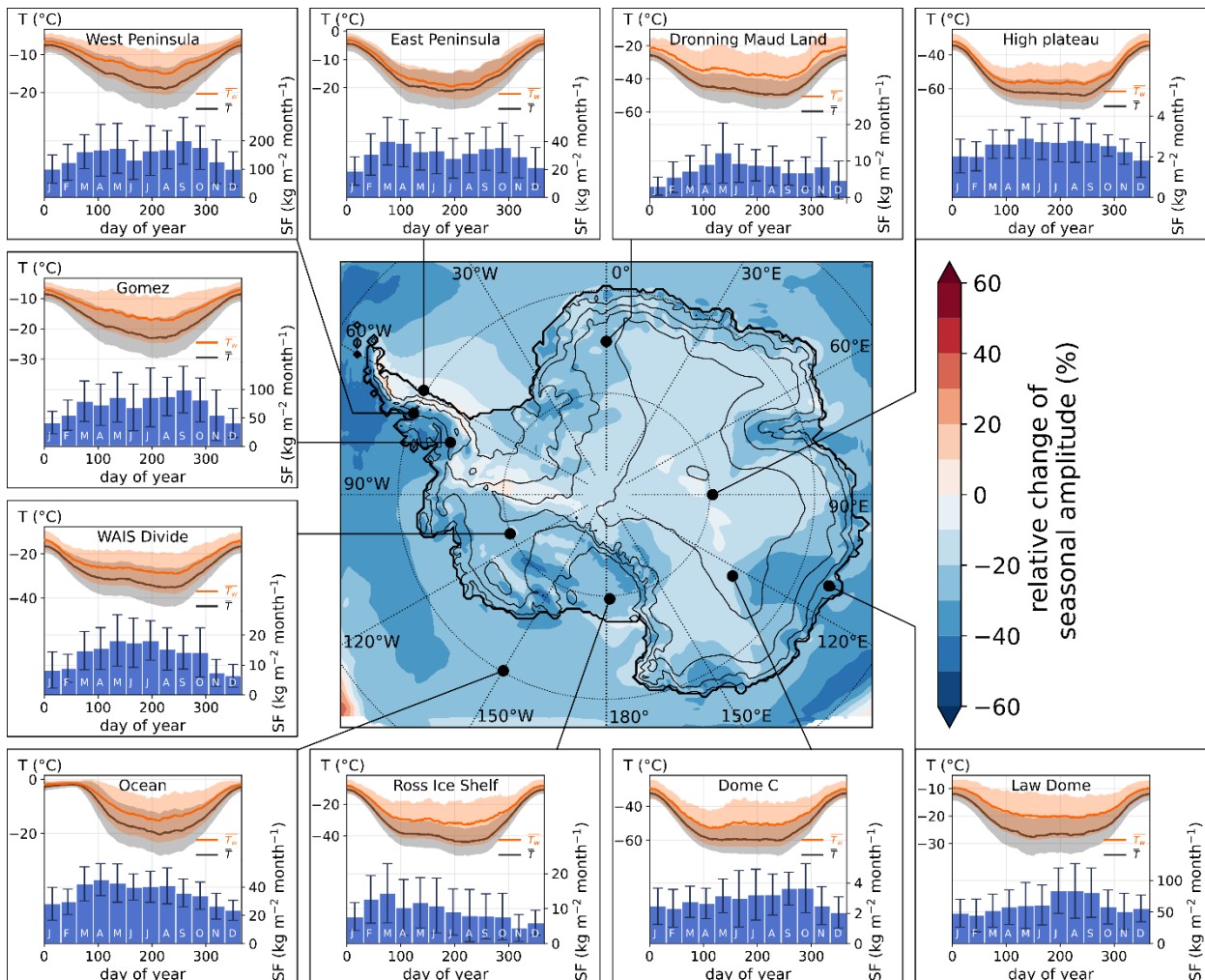

**Figure 5.** Map of the relative change in seasonal amplitude, defined as the ratio of standard deviations $r_{SD} = \left(\frac{SD(\overline{T_w})}{SD(\overline{T})} - 1\right) \times 100$ for each model cell, where SD is the standard deviation. Inserts show seasonal cycles of temperature and snowfall for a selection of 10 sites in Antarctica. For each site, climate normals of temperature ($\overline{T}$), snowfall-weighted temperature ($\overline{T_w}$) and monthly snowfall, similarly to Fig. 4. Climate normal temperatures were computed as arithmetic (for $\overline{T}$) or snowfall-weighted (for $\overline{T_w}$) average of the same day of year for all the 1979–2020 period, then smoothed with a 30-days rolling mean.

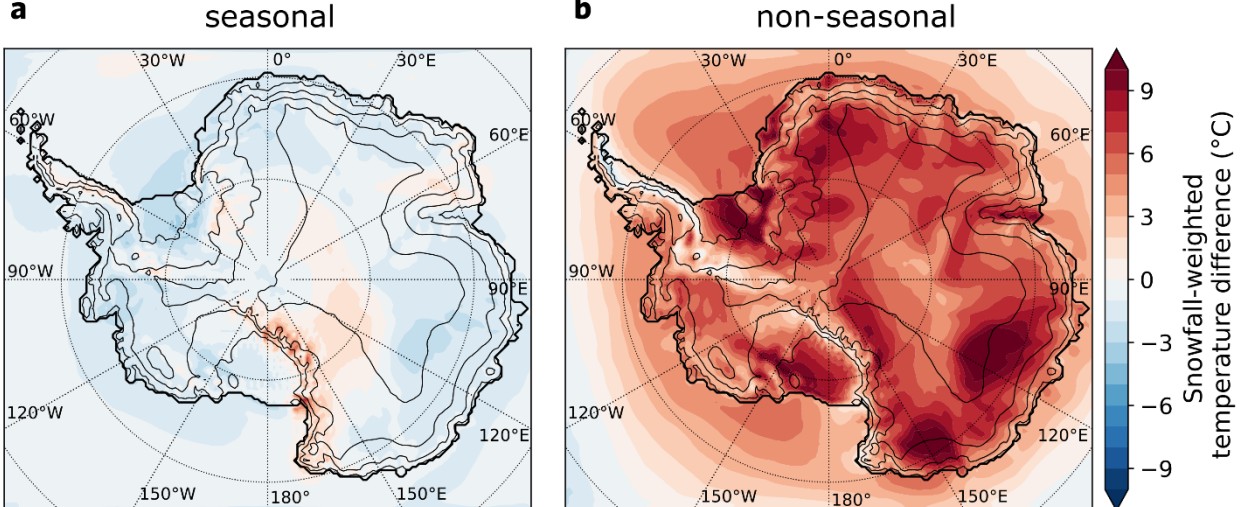

**Figure 6.** Seasonal and non-seasonal effects of snowfall weighting on temperature difference (ΔT). (a) seasonal effect on temperature difference reflects the temperature changes induced by snowfall seasonality. (b) non-seasonal effects of snowfall weighting, revealing the influence of snowfall event-related warming. The sum of both maps results in the map shown in Fig. 2 (see Sect. 2 methods).

**Figure 7.** Maps of the change of interannual standard deviation of temperature induced by the snowfall weighting. **(a)** standard deviation of yearly averaged temperature $SD(^yT)$. **(b)** standard deviation of annually snowfall-weighted temperature $SD(^yT_w)$. **(c)** difference of standard deviations $SD(^yT_w) - SD(^yT)$. **(d)** relative change of standard deviation, given by $r_{SD} = \left(\frac{SD(^yT_w)}{SD(^yT)} - 1\right) \times 100$. For all definitions, $SD$ is the standard deviation, $^yT_w$ and $^yT$ are yearly snowfall-weighted temperature and yearly average temperature, respectively. Thick black lines indicate the extent of the Antarctic continent (including ice shelves). Thin lines show the 100 m, 1000 m, 2000 m, and 3000 m elevation contours.

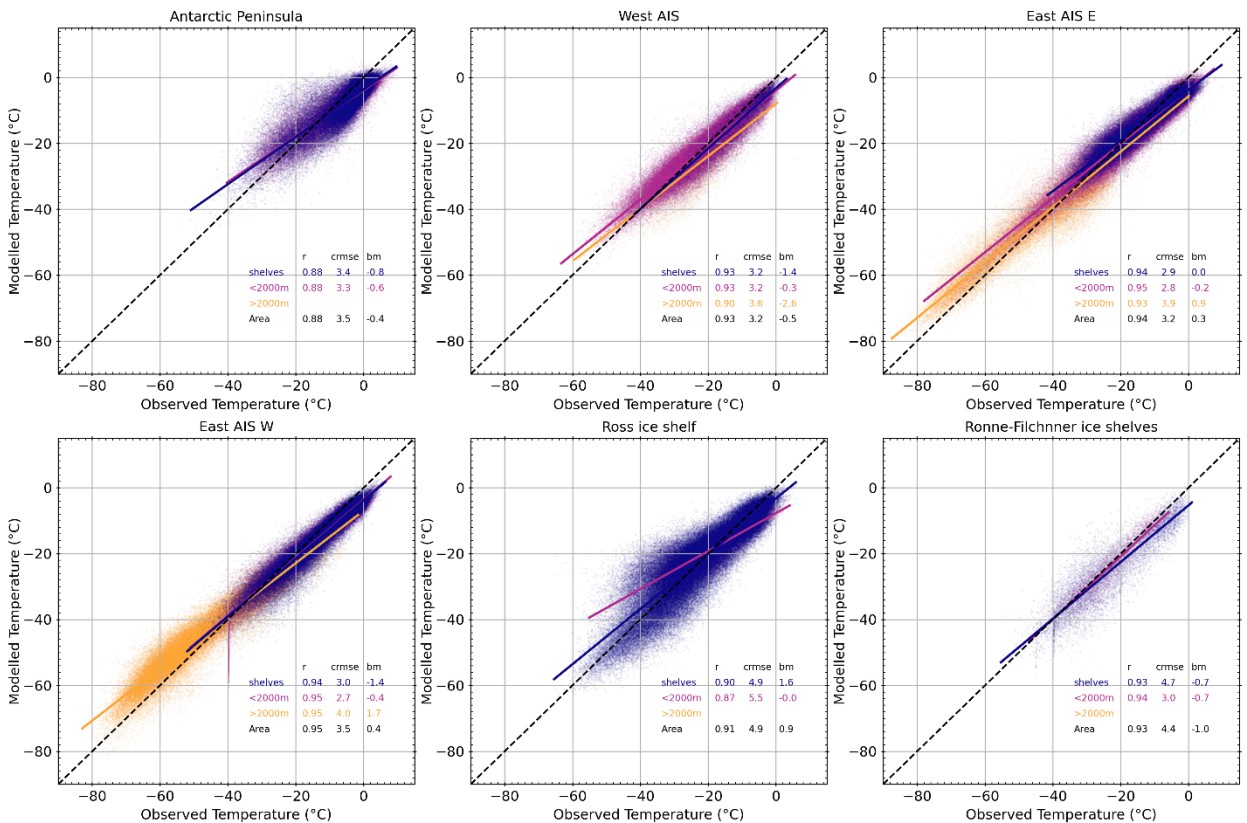

**Figure A1.** Evaluation of model performance to represent temperature. Scatterplots and linear regressions of modelled (MAR v3.12) vs observed temperature (compilation of automatic weather stations and other measurements Mottram et al., 2021) for different regions (as in Kittel et al., 2021). Slopes slightly lower than 1 indicate that the natural range of temperature variability is greater than what the model can achieve, although the difference is minor. Residual Mean Square Error (RMSE) measures the scattering of modelled temperatures around the observed temperature, and Bias Mean (BM) measures the average difference with the observed temperature. Both give an estimation of imprecision of the model.

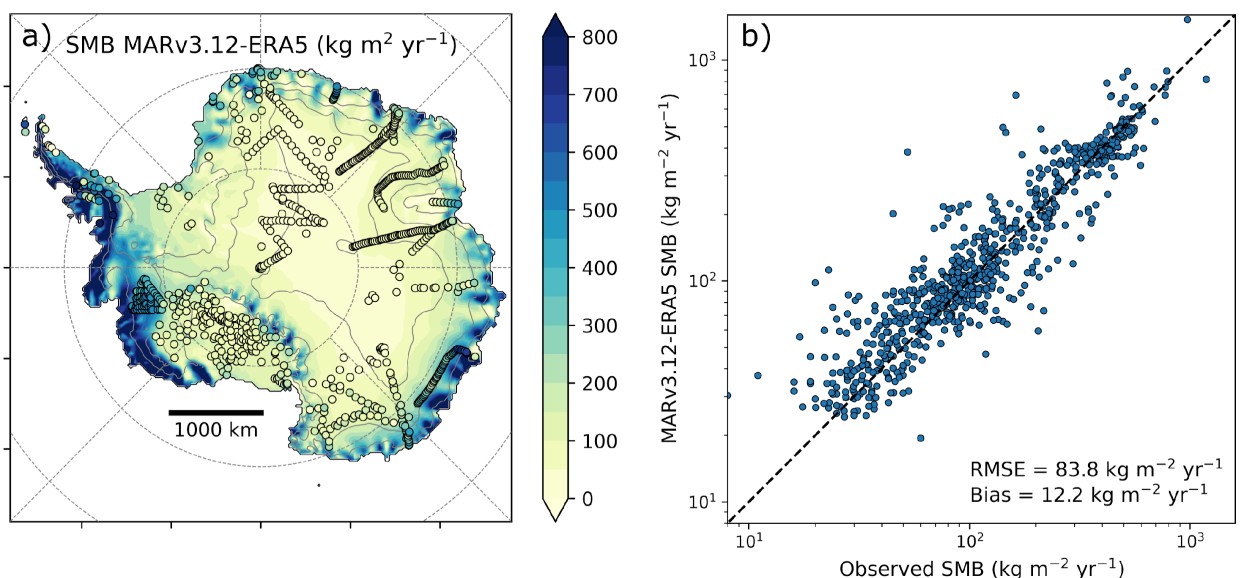

**Figure A2.** Evaluation of model performance to represent Surface Mass Balance. (**a**) map of Surface Mass Balance (SMB), defined as precipitation minus evaporation and runoff in MARv2.13, compared to accumulation observations (Wang et al., 2016) represented as colour dots. (**b**) log-scale scatterplot of modelled SMB vs accumulation observations. Modelled SMB is higher than observations by a bias of 12.2 kg m$^{-2}$ yr$^{-1}$.

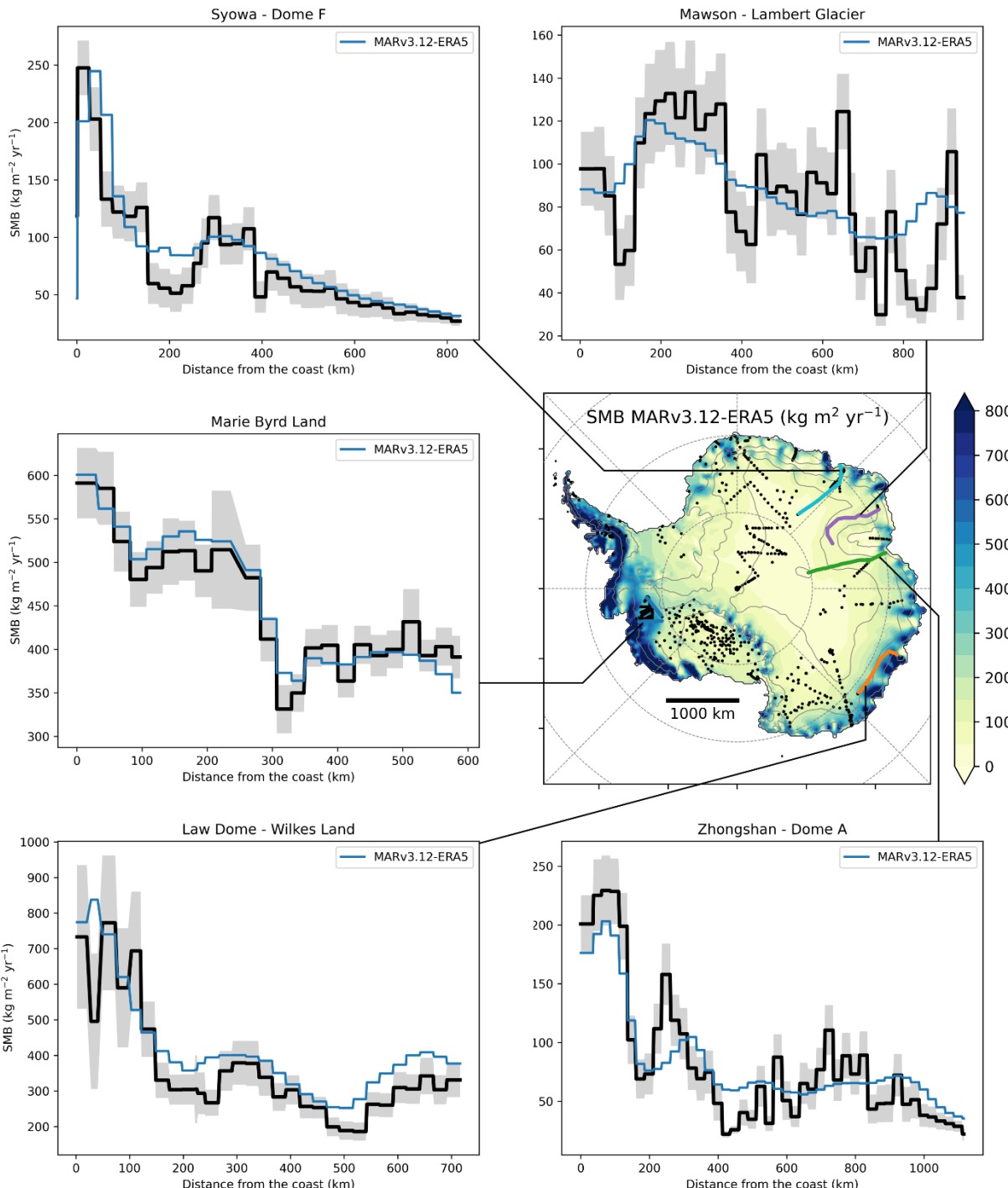

**Figure A3.** Evaluation of Surface Mass Balance model performance along transects Surface Mass Balance (SMB), defined as precipitation minus evaporation and runoff in MARv2.13 is compared to observations (Wang et al., 2016) along five transects represented by colour lines on the map. MAR tends to slightly overestimate SMB at high elevation sites, and underestimate the variability of small-scale changes. This

could be attributed to unresolved drifting snow transport in MAR due to coarser than real topography (Agosta et al., 2019). Despite these flaws, modelled SMB remarkably follows the observed spatial trends in accumulation.

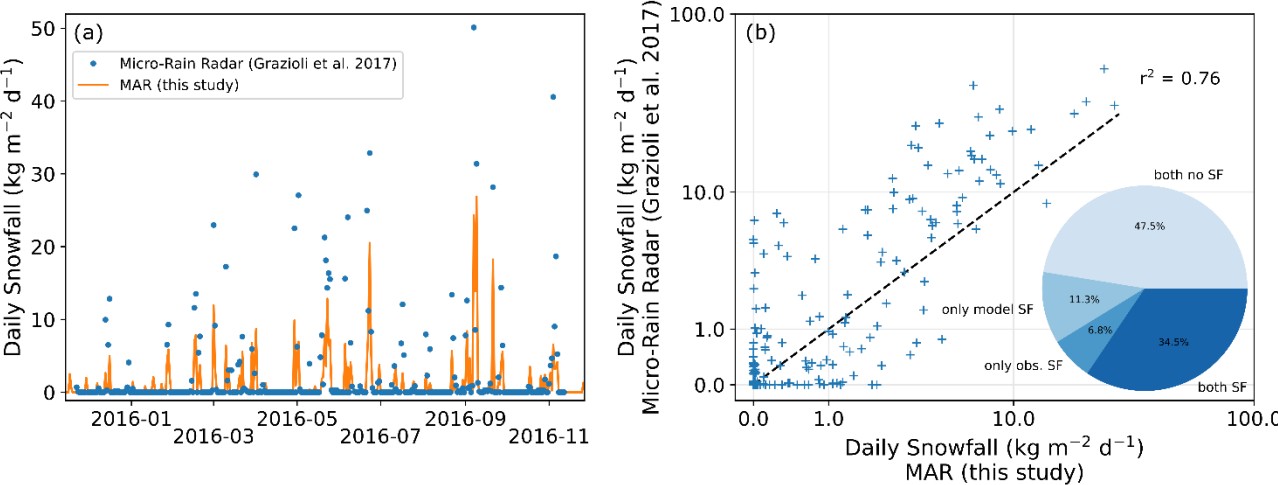

**Figure A4.** Evaluation of MAR to match snowfall timing observed with micro-rain radar at Dumont D'Urville station (66 °S, 140 °E, Grazioli et al., 2017). Micro-rain radar data indicates snow passing through the atmospheric layer 300 m above the surface, while modelled snowfall is shown for the surface only, where some of the snow may have been sublimated. (a) time-series of modelled and observed snowfall for the year 2016. (b) scatter plot of observed vs modelled snowfall. The Pie-chart indicates the percentage of days with matching or mismatching snowfall in both model and observations, with discrepancies noted for about 18 % of days in total.

615