# Peer review of "Control of the temperature signal in Antarctic proxies by snowfall dynamics"

_EGUsphere, 2023_

## Author Comment (AC1)

[Figure]

Figure A4. Evaluation of MAR to match snowfall timing observed with micro-rain radar at Dumont D'Urville station (66 °S, 140 °E, Grazioli et al. 2017). Micro-rain radar data indicates snow passing through the atmospheric layer 300 m above the surface, while modelled snowfall is shown for the surface only, where some of the snow may have been sublimated. (a) time-series of modelled and observed snowfall for the year 2016. (b) scatter plot of observed vs modelled snowfall. The Pie-chart indicates the percentage of days with matching or mis-matching snowfall in both model and observations, with discrepancies noted for about 18 % of days in total.

[Figure]

[Figure]

---

## Author Comment (AC3)

[Figure]

Figure 3. Scatter heatmap of annual snowfall -weighted temperature ($T_w$) as a function of annual temperature ($T$) for every model point on the surface of the Antarctic ice sheet (including ice shelves). The red continuous line represents the average snowfall-weighted temperature given the annual temperature, dashed line highlights 1:1 line

[Figure]

Reply to Review #3, Supporting Figure.
Decomposition of DeltaT using frequency filters: highpass (cut-off 60 days), bandpass (cut-offs 60 to 375 days) and lowpass (cut-off 375 days) to respectively represent the synoptic, seasonal and interannual effects of snowfall-weighting, as in Sime et al. (2008)
(a) example of the time-series filtered signals for Dome C.
(b) map of low-passed DeltaT
(c) map of band-passed DeltaT
(d) map of high-passed DeltatT

---

## Author Response (AR2)

**Reply on Editor's Decision on Minor revision by Tas van Ommen**

Comments from the Editor to which we reply directly are copied here *in italic*. Our response follows, and modifications to the manuscript are highlighted in bullet points.

*Line 54: suggest that 'precipitation' (singular) is best for one type even where plurality is intended – also alter verb '…precipitation forms…' and also line 55 – if the authors wish to stress the collective ensemble of precipitation events or multiple types (clear-sky and cloud-originating precipitations) in a sentence then precipitations (plural) is acceptable, but this seems unnecessary here. This recurs in the MS (e.g. lines 70, 278, 295) and should be altered throughout.*

We changed to use precipitation (singular).

*Line 81: the words "a few large precipitation events" primarily describes the for the inland plateau and is not so applicable to the coastal higher accumulation sites. Better to say, "Particularly in inland Antarctica, at higher elevations, a few large precipitation events…". This also helps clarify the subsequent sentence, as "the few days" really doesn't describe higher accumulation coastal zones as well.*

The formulation made it sound inexact; we revised the sentence to describe more closely the results of the cited paper. Please note that these results are not restricted to inland antarctica, and high-accumulation coastal regions are particularly affected by the precipitation variability, as described by Turner et al. (2019).

- In Antarctica, the 10 % largest daily precipitation events at a location can cause most of the accumulation annual amount and interannual variability (Turner et al., 2019)

*Line 92: 'non-related' is better as unrelated or not related*

Corrected

*Line 173: "…higher than [the] average temperature…"*

Corrected

*Line 285: "lower annual cycle" is imprecise – do you mean "attenuated", or "smaller"?*

Changed to "may reflect an annual cycle of $\delta^{18}O$ with attenuated amplitude"

*Figure 3: tidy the right axis label on top panel, part b, e.g. by cropping red mean Tw line*

It should not appear in the final figure files, uploaded separately.

**Reply on Editor's Decision on Interactive discussion by Tas van Ommen**

Comments from the Editor to which we reply directly are copied here *in italic*. Our response follows, and modifications to the manuscript are highlighted in bullet points.

*Additional private note (visible to authors and reviewers only):*
*Your proposed bullet point addressing the condensation temperature versus 2m temperature issue covers the issue adequately. The reader will, however, be interested in how this additional "transfer function" impacts the interpretation of the water isotope thermometer. You go some way to commenting on this, regarding the impact of temperature inversion on the seasonal cycle:*
*"With condensation temperature, we would expect weaker seasonal cycles because winter surface cooling is amplified by a strong inversion, but long-term temperature variability may not change much as implied by deglaciation simulations (Liu et al., 2023)..."*
*This is a good point to make - do you feel able to make any additional comments around, for example, the expected impact on the strength of the correlation between temperature and the snowfall weighted temperature at condensation level which is imprinted on the isotopes?*

We expect that using condensation level has at least two opposite effects on the correlation of temperature with snowfall-weighted temperature.

1/ correlation could be increased when looking at the condensation level where it is more likely that temperature and precipitation covary: temperature and precipitation are directly linked by the presence of precipitating cloud, in an air mass originating form warmer location, increasing the long wave absorbance, and possibly releasing latent heat associated with condensation.

2/ However, condensation level may change vertically (Durán-Alarcón et al. 2019 doi:10.5194/tc-13-247-2019, Gorodetskaya et al. 2015 doi:10.5194/tc-9-285-2015), even at the timescale of a single precipitation event with the passage of atmospheric front, inducing variability in the condensation temperature, and its difference with average temperature, because the temperature decreases with altitude. This would introduce a lot of noise in the snowfall-weighted condensation temperature and decrease correlation.

Of these two effects, we cannot estimate which would be dominant without making the calculations. This would require substantial amount of work because it implies to use all atmospheric levels in the model, as opposed to only the surface parameters used in the submitted manuscript. Linking condensation temperature (and precipitation weighted temperatures) to surface temperature in the MAR model deserves an evaluation of its own, in a different work. It should compare to studies directly linking water isotopes to the surface temperature.

In this manuscript we primarily discussed how the snowfall deposition occurs under anormal temperature conditions, and how it will affect the values of isotope (or snowfall-weighted temperature) – temperature slope. We did not provide numbers on how the correlation could be reduced between these two parameters. I think this topic has been treated quite well in other papers (Sime et al. 2009b, Casado et al. 2020, both cited in the manuscript, and Münch and Laepple 2018 doi:10.5194/cp-14-2053-2018 that we did not cite).

Although we could study of the correlation loss due to snowfall weighting and snowfall intermittency, it would require new figures and text, as the effect depends on the time window of the average. At this point it may be too late to add so much new content to the current manuscript.

Because we did not discuss correlation in detail, we prefer not to add a speculative note on how using condensation level may modify the correlation. We will however keep these discussion points in mind so we can address them in a potential future study.

**Reply on RC1 by Anonymous Referee #1**

Comments from the Referee to which we reply directly are copied here *in italic*. Our response follows, and modifications to the manuscript are highlighted in bullet points.

*Major comments*

*In the introduction, it could be made more clear that it is the temperature at the elevation of precipitation formation (the condensation temperature) that is imprinted in the snow, and not near-surface or surface temperature. This temperature is then often regressed onto average surface temperature (from 10 m snow temperatures) to make the coupling of the isotopic signal to the surface.*

We detailed that condensation temperature is the most important in the intro (sect 1):

> Due to Rayleigh distillation during transport of moisture to cold regions, water isotopes reflect the condensation temperature of precipitations (Dansgaard, 1964). However, the relationship between average temperature at a location and isotopes in the snow is altered by deposition dynamics of snowfall-born water isotopes (Werner et al., 2000; Persson et al., 2011; Casado et al., 2020), post-deposition processes such as ablation-redeposition and sublimation-condensation cycles (Steen-Larsen et al., 2014; Touzeau et al., 2016; Stenni et al., 2016; Münch et al., 2017; Hughes et al., 2021), and the difference between condensation and surface temperature (Buizert et al., 2021; Liu et al., 2023).

And justified the use of surface temperature in method (sect 2.2):

> Although the temperature recorded in water isotopes is imprinted at the condensation level (Jouzel and Merlivat, 1984), we chose to use 2-m air temperature for simplicity, because condensation levels change both spatially and temporally. Studies using water isotopes usually bypass the condensation to surface temperature changes by directly calibrating the isotope-temperature slope with 2-m temperature in most cases (e.g., Jouzel et al., 2007; Stenni et al., 2017), or applying a ratio of temperature changes that would be amplified at the surface (e.g., Jouzel et al., 2003). If we used the condensation-level temperature, the difference with climate normal would depend on the level of precipitation formation, and may be vertically spread on the atmospheric column, making the comparison more complex. With condensation temperature, we would expect weaker seasonal cycles because winter surface cooling is amplified by a strong inversion, but long-term temperature variability may not change much as implied by deglaciation simulations (Liu et al., 2023). Choosing the surface temperature also enables comparison with available observations, and this is the level also considered in many paleotemperature reconstructions.

*l. 69: "extensively evaluated for its representation of Antarctic surface mass balance and temperature". This is true, but e.g. Mottram and others (2022) show that MAR3.10 appears to be significantly above-average wet in the East Antarctic region west of the Ross ice shelf, also one of the delta_T hotspots in Fig. 2. Moreover, the model is not evaluated for the key variables used in this paper, i.e., the timing of precipitation. Any comments?*

I suppose you refer to Mottram et al. (2021). Compared to observations, MAR overestimates the Surface Mass Balance (SMB) on the Ross ice shelf, whereas the hotspot of $\Delta T$ is on grounded ice West of Ross ice shelf, were there are no SMB observations due to SMB being so small altogether in this region. The few

exceptional snowfall events that reach this region can therefore differ substantially from the average cold conditions. Due to the lack of observations to confirm the SMB or precipitations we prefer not to write this speculative guess in the manuscript. For the Ross ice shelf, seasonal misdistribution may affect the seasonal effect on $\Delta T$, which is currently relatively neutral. This potential bias should be a subject of exploration in future SMB evaluations, for all regions.

Regarding the timing of precipitations, little observations are available. Now that more instruments capable of evaluating snowfall have been deployed on the field, future model evaluations may also be compared to the produced observations. Of the few published works, we have been able to compare the timing of precipitation to a micro-rain radar derived snowfall dataset (Grazioli et al., 2017) for only one location and one year, and will include it in the Appendix as an evaluation of precipitation timing (also attached to this reply).

*Figure 3: Consider including standard deviation in the temperature curves and precipitation bars, to indicate the temporal variability on which these averages are based. This also supports the statement about temperature variability in winter in l. 154.*

We revised the figures and respective captions to include standard deviation shading and error-bars. Please see the revised figures attached.

***Minor and textual comments***

*Please use 'higher' and 'lower' temperatures rather than 'warmer' and 'colder/cooler' temperatures throughout; I realize it is a rearguard battle but hey, that is the privilege of the reviewer!*

We changed the text where warm/cool and temperature were used in the same sentence.

*l. 38: "ablation-redeposition and sublimation-condensation" These combinations are not necessarily mutually exclusive. Did you mean "erosion/sublimation and deposition cycles"?*

This formulation intended to emphasize the difference between macro- and micro-physics, with mixing of snow by the wind (ablation-redeposition) at macro-scale and molecular diffusion (sublimation-condensation) cycles. Both are **post-deposition** processes that are acknowledged in the introduction, but are not treated in this manuscript, which focuses only on the initial deposition (first half of the sentence: "between average temperature at a location and isotopes in the snow is altered by deposition dynamics of snowfall-borne water isotopes").

*l. 151: This formulation could be condensed to: " emerges from the stronger near-surface horizontal and vertical temperature gradients..."*

The variation is not only spatial, but temporal in that case, so the suggested reformulation is not suited. Nonetheless, we rewrote this sentence to make it easier to read:

> The larger difference in winter results from the attenuation of near-surface temperature inversion during the passage of precipitating atmospheric systems.

*l. 202: "snowfall-weighted δ18O " Do you mean oxygen isotopes in atmospheric water vapor? Please clarify.*

Added "of precipitations"

*l. 223: "Snowfall-weighted climate normal temperature " This is unclear, please reformulate or clarify.*

replaced with "seasonal cycle of temperature during snowfall"

**references**

[revised manuscript text omitted]

**Revised figures including standard deviations:**

[Figure]

**Reply on RC2 by Anonymous Referee #2**

Comments from the Referee to which we reply directly are copied here *in italic*. Our response follows, and modifications to the manuscript are highlighted in bullet points.

*Major comments (but minor revisions)*

*Stable isotopes of water are mentioned in the article from the second sentence and throughout the rest of the paragraph, with more detailed descriptions of the processes controlling isotopic signals in Antarctic firn and ice cores. I think it's a little bit too harsh and too specific considering the main topic of this paper, even if the findings of this study have important implications for the paleoclimate reconstructions using stable water isotopes in Antarctic ice cores. To make it simple, I think the two first paragraphs could be swapped (with some adaptation). Moreover, it would make a smoother transition with the 3rd paragraph.*

The two paragraphs have been swapped, and we added a short general phrase to start the paragraph:

> Antarctica is the coldest and driest continent on earth, and almost entirely covered by ice. The surface temperature remains below freezing year-round over most of the continent, allowing the snow to accumulate and form the ice sheet, which is recharged primarily by snowfall. Precipitating atmospheric systems in polar regions (…)

*One of the most interesting findings concerns the greater inter-annual variability of snowfall-weighted temperature compared with annual temperature. Could you try to establish a link with an index of internal climate variability such as the Southern Annular Mode (SAM)? For example, Kino et al (2021) have shown the impact of SAM on the water isotope temperature record at Fuji Dome, through changes in atmospheric circulation.*

Previous studies highlighted changes in temperature and precipitation specifically related to SAM in most of Antarctica (Marshall and Thompson, 2016; Marshall et al., 2017). We also find a weak but significant negative correlation between temperature and SAM in most of Antarctica, except for peninsula (Supporting Figure 1), as highlighted in the cited studies. A brief evaluation of SAM impact on snowfall weighted temperatures (Supporting Figure 2) shows no correlation between SAM and the DeltaT at monthly scale. We prefer not to discuss this topic in detail in the current manuscript, but include a brief mention in the discussion (Section 3.3):

> Links were found between Antarctic temperature and large-scale atmospheric circulation patterns in the Southern Hemisphere such as the southern annular mode (Marshall and Thompson, 2016), possibly influencing the $\delta^{18}O$ of ice cores (Abram et al., 2014; Kino et al., 2021). Nevertheless, we did not find any significant correlation between the SAM and yearly or monthly snowfall-weighted temperature difference. Detecting possible links between the SAM, or other climate modes, and the precipitation-weighted temperature (or $\delta^{18}O$) would require a more detailed investigation, and may be explored in a different study.

*2m air temperature is used for analysis. Could you explain in a few sentences the differences you would expect if condensation temperature were used instead?*

We modified the paragraph justifying the use of 2-m temperature:

Although the temperature recorded in water isotopes is imprinted at the condensation level (Jouzel and Merlivat, 1984), we chose to use 2-m air temperature for simplicity, because condensation levels change both spatially and temporally. Studies using water isotopes usually bypass the condensation to surface temperature changes by directly calibrating the isotope-temperature slope with 2-m temperature in most cases (e.g., Jouzel et al., 2007; Stenni et al., 2017), or applying a ratio of temperature changes that would be amplified at the surface (e.g., Jouzel et al., 2003). If we used the condensation-level temperature, the difference with climate normal would depend on the level of precipitation formation, and may be vertically spread on the atmospheric column, making the comparison more complex. With condensation temperature, we would expect weaker seasonal cycles because winter surface cooling is amplified by a strong inversion, but long-term temperature variability may not change much as implied by deglaciation simulations (Liu et al., 2023). Choosing the surface temperature also enables comparison with available observations, and this is the level also considered in many paleotemperature reconstructions.

*Minor technical comments:*

*Line 27: reduced by 20% compared to what?*

Rephrased to:

Temperature during snowfall has a seasonal amplitude reduced by 20 % relative to the daily temperature.

*Line 44: "are known to increase the surface temperature". I agree with the comment of the first reviewer about higher and lower temperatures (and not warmer and cooler temperatures).*

We changed the text where warm/cool and temperature were used in the same sentence.

*Line 82: which fields of MAR are nudged to ERA5 (U and V winds?)? Please give some more details. Moreover, the proper reference to ERA5 reanalyses is Hersbach et al. (2020).*

Added in methods (Section 2.1):

MAR is forced with 6-hourly outputs of the ERA5 TL95 reanalysis (Hersbach et al., 2020) at its lateral boundaries (temperature, wind, humidity) and for upper-air relaxation at the top of the troposphere (temperature, wind), and with daily outputs at the surface of the ocean (sea surface temperature, sea ice concentration).

*Lines 188-191: Other studies before weighted the d18O and temperature by daily variations of precipitation (and not by monthly variations only) to study the isotope-temperature temporal relationships, like in Werner et al. (2018).*

The suggested article mainly discusses the effect of topography on the isotope – temperature slope, and differences in spatial vs temporal slopes. It also suggests that "reconstructions of precipitation-weighted mean temperatures" are more suited from isotopes, although here in our manuscript we try to tackle this problem by looking at the difference between precipitation-weighted mean temperatures and "true" mean temperatures. Therefore, we did not find enough similarities to compare our results with. We however added a reference to the article at the relevant place in the introduction:

[revised manuscript text omitted]

Werner, M., Jouzel, J., Masson-Delmotte, V., and Lohmann, G.: Reconciling glacial Antarctic water stable isotopes with ice sheet topography and the isotopic paleothermometer, Nat Commun, 9, 3537, https://doi.org/10.1038/s41467-018-05430-y, 2018.

**Supporting Figures**

[Figure]

Supporting Figure 1, Reply to Review #2:

Scatter-plots and correlation values for monthly SAM index and monthly temperature anomalies, at ten sites described in the manuscript. Most sites show a weak negative correlation with SAM index, except for the Peninsula sites (incuding Gomez) where no correlation to weak positive correlation was found.

SOURCE for SAM monthly index:
https://legacy.bas.ac.uk/met/gjma/sam.html [accessed 2023-10-13]

[Figure]

Supporting Figure 2, Reply to Review #2:

Scatter-plots and correlation values for monthly SAM index and monthly snowfall-weighted minus average temperature, at ten sites described in the manuscript. Most sites show no correlation with SAM index, suggesting that SAM does not directly affect the snowfall-weighting difference of temperature averaging.

SOURCE for SAM monthly index:
https://legacy.bas.ac.uk/met/gjma/sam.html [accessed 2023-10-13]

**Reply on RC3 by Anonymous Referee #3**

Comments from the Referee to which we reply directly are copied here *in italic*. Our response follows, and modifications to the manuscript are highlighted in bullet points.

*Major comment (i) The missing literature on precipitation intermittency:*

Given the similarity of the suggested works with the current study, we have no excuse for missing out on these papers. We therefore thank the reviewer for the recommendations that will greatly enrich the manuscript, and made the necessary changes.

Before detailing the changes, reading this bibliography inspired us a new figure, which is relevant for sections 3.1 and 3.3. We added this descriptive text in section 3.1, and referred to it again in the revised section 3.3. Figure numbers in revised text therefore reflect the addition of this new Figure (Fig. 3), and are re-numbered accordingly (Figs 3-6 in the original manuscript are now Figs 4-7). Note that we do not talk about isotopes in section 3.1, therefore do not refer to (Sime et al., 2009a) in this paragraph, but do cite their work in section 3.3.

> The analysis of yearly snowfall-weighted temperature ($^yT_w$) and "true" yearly temperature ($^yT$, Fig. 3) further supports that the effect of snowfall weighting is not constant, and may differ along local parameters including the temperature, but also probably the precipitation regimes. Importantly, $^yT_w$ is not linear with $^yT$, suggesting that changes in the annual temperature are not matched by proportional changes in the snowfall-weighted temperature. This relationship may also change whether we average annually or at other time resolutions. Besides, any given $^yT$ is matched by a large distribution of $^yT_w$, which means that snowfall weighting induces variability in the temperature.

**Detailed changes to include references to suggested bibliography.**

We added a paragraph in the introduction to refer the previous studies and their general findings how the objectives of the present manuscript may complete them:

At the end of the first introduction paragraph:

[revised manuscript text omitted]

Now, applying the frequency decomposition method as in Sime et al. (2008) is possible. In the current manuscript we opted for a decomposition onto climate normal + anomaly, as opposed to frequency-filtering the temperature and precipitation used for bias. We made the maps of temperature difference using the decomposition method described in Sime et al. (2008), in the supporting figure attached. The interannual $\Delta T$ computed with a lowpass is consistent with Sime et al. (2008) who describe a <|0.5°C| bias at interannual scale; this means that most of the remaining signal is split into seasonal (60 to 375 days band-pass) and synoptic (60 days high-pass) scales, and yields similar results as we described in the manuscript (Figure 5, renamed to figure 6 in the revised manuscript, see the discussion above in this reply for the additional figure). Due to the low interannual bias, the two methods are approximately equivalent.

We chose to continue using our decomposition as the distribution of precipitation throughout the year is often a topic of discussion for seasonal biases, so using the convolution of precipitation along the climate normal temperature is more direct for this specific discussion. In particular, deviation from this climate normal temperature, namely temperature anomaly (T'), is the variable shown in Fig.1 and in the inserts in Fig. 2. In addition, we show the climate normal temperature in Figs. 3 and 4 (Figs. 4 and 5 in the revised manuscript), thus we prefer to keep the consistency between current figures.

Finally, we made additions in Section 3.3 to include suggested papers:

In second paragraph of 3.3:

> Previous studies also highlighted that despite being weaker that non-seasonal effects in absolute value, seasonal effects on $\Delta T$ are the more likely to vary with climate as the seasonality of precipitation changes (Sime et al., 2008), in response to sea ice and moisture source changes (Holloway et al., 2016).
>
> […]
>
> Given the spatial variability of $\Delta T$, we advise against the use of spatial gradients to define isotope-temperature slopes for temporal reconstructions.

After third paragraph of 3.3:

> This explains at least partly a higher interannual variability of precipitation-weighted $\delta^{18}O$, causing increased $\delta^{18}O$-temperature slope in most of Antarctica at interannual scale compared to seasonal scale (Goursaud et al., 2018), and low correlations between modelled $\delta^{18}O$ and temperature at annual scale (Münch et al., 2021). Simulation of $\delta^{18}O$ signals that would be recorded in Antarctic Peninsula ice cores also revealed that the interannual variability in $\delta^{18}O$ may show poor correlation to temperature variability even in high accumulation regions (Sime et al., 2009b). Non-linearities in the snowfall-weighted temperature as temperature and climate changes (Fig. 3) may be responsible for non-linear response of isotopes to temperature and underestimation of temperature maximum in warm periods, through increased winter (Sime et al., 2009a).

Revised final paragraphs of 3.3:

[…] Moreover, using slopes variable through time would result in better temperature quantification, because the slope depends on the temperature range and the location (Sime et al., 2009a), and may vary through time (Klein et al., 2019).

Quantifying the local effect of snowfall-weighting on temperature range can help refine the temperature-isotope slopes for a more accurate estimation, and it should be done for different settings from glacial to warmer-than-present interglacial climate. Future temperature reconstructions could consider proceeding in two steps: (1) determine the snowfall-weighted temperature from water isotopes, for which the correlation is generally good and can be determined by Rayleigh-type models (e.g., Markle and Steig, 2022), then (2) determine the average (non-weighted) temperature through site-calibrated $T_w - T$ slope, calculated for the matching temporal resolution (similarly to Fig. 3, but here we only show the $^yT_w - {}^yT$ slope computed with yearly averages, and include all of Antarctica), while accounting for the difference in temperature between condensation level and surface, often dictated by inversion strength. Greater snowfall-weighted temperature differences at low-accumulation sites suggest that changes in snowfall regimes could impact the temperature difference, and thus bias the reconstructions from isotopes. Further work is necessary to fully understand how change in snowfall dynamics may influence temperature reconstructions from isotopes, which may be facilitated by atmospheric models equipped with isotopes.

Unfortunately, despite our effort to search cross-referenced papers, not many other works have relevance for the specific topic of how precipitation weighting may affect the temperature signal. We added a few references in introduction and in Section 3.3 (Krinner et al., 2006; Goursaud et al., 2018; Klein et al., 2019; Münch et al., 2021; detailed changes above).

*Major comment (ii) The importance of surface versus condensation temperature:*

2.2 first paragraph was further detailed:

Although the temperature recorded in water isotopes is imprinted at the condensation level (Jouzel and Merlivat, 1984), we chose to use 2-m air temperature for simplicity, because condensation levels change both spatially and temporally. Studies using water isotopes usually bypass the condensation to surface temperature changes by directly calibrating the isotope-temperature slope with 2-m temperature in most cases (e.g., Jouzel et al., 2007; Stenni et al., 2017), or applying a ratio of temperature changes that would be amplified at the surface (e.g., Jouzel et al., 2003). If we used the condensation-level temperature, the difference with climate normal would depend on the level of precipitation formation, and may be vertically spread on the atmospheric column, making the comparison more complex. With condensation temperature, we would expect weaker seasonal cycles because winter surface cooling is amplified by a strong inversion, but long-term temperature variability may not change much as implied by deglaciation simulations (Liu et al., 2023). Choosing the surface temperature also enables comparison with available observations, and this is the level also considered in many paleotemperature reconstructions.

*Minor comments:*

*Introduction – needs to be fairly substantially modified in the light of the above.*

A new paragraph was added to highlight previous similar works (see additions above). Moreover, as suggested by the Review #2, we re-ordered the introduction so that isotopes are now mentioned from the second paragraph, with the first paragraph focusing on the warming effects of precipitations, the main topic of the first half of this manuscript.

*Line 124 – please compare with the equivalent numbers from previous HadCM3 and ERA40 results in the 2008 and 2011 papers.*

See additions above, the comparison is made throughout Section 3.

*Line 167 – add calculations also for the inter-annual terms using MAR-ERA5 output.*

Detailed calculations are now written in the figure caption, along with the yearly averaged variables noted $^yT$ and $^yT_w$, used for the new Fig. 3 and added to Table 1.

*3.3 needs quite a lot of rewriting to acknowledge that whilst previous authors have calculated the daily biasing effects – and have shown these to be largest - nevertheless the most terms that changes the most with climate is generally the seasonal, rather than the daily/synoptic biasing terms. On this, do also read and consider referencing: Holloway, Max D. , Sime, Louise C. , Singarayer, Joy S., Tindall, Julia C., Bunch, Pete, Valdes, Paul J.. (2016) Antarctic last interglacial isotope peak in response to sea ice retreat not ice-sheet collapse. Nature Communications, 7. 9 pp. doi:10.1038/ncomms12293. Text can be modified to reflect that this paper also shows the primacy of seasonal (change with climate) effects. The 2008, 2009 and 2011 papers, noted above, methods and results should also accounted for during rewriting.*

See additions above, the suggested article was cited in section 3.3.

**References**

[revised manuscript text omitted]

Reply to Review #3, Supporting Figure.
Decomposition of DeltaT using frequency filters: highpass (cut-off 60 days), bandpass (cut-offs 60 to 375 days) and lowpass (cut-off 375 days)
to respectively represent the synoptic, seasonal and interannual effects of snowfall-weighting, as in Sime et al. (2008)
(a) example of the time-series filtered signals for Dome C.
(b) map of low-passed DeltaT
(c) map of band-passed DeltaT
(d) map of high-passed DeltatT